# Importance of Cardiopulmonary Exercise Testing amongst Subjects Recovering from COVID-19

**DOI:** 10.3390/diagnostics11030507

**Published:** 2021-03-12

**Authors:** Gianluigi Dorelli, Michele Braggio, Daniele Gabbiani, Fabiana Busti, Marco Caminati, Gianenrico Senna, Domenico Girelli, Pierantonio Laveneziana, Marcello Ferrari, Giulia Sartori, Luca Dalle Carbonare, Ernesto Crisafulli

**Affiliations:** 1School of Medicine in Sports and Exercise, University of Verona, 37134 Verona, Italy; gianluigi.dorelli@gmail.com (G.D.); michele.braggio@gmail.com (M.B.); marcello.ferrari@univr.it (M.F.); luca.dallecarbonare@univr.it (L.D.C.); 2Department of Medicine, Section of Internal Medicine, University of Verona and Azienda Ospedaliera Universitaria Integrata of Verona, 37134 Verona, Italy; daniele.gabbiani@yahoo.com (D.G.); fabiana.busti@univr.it (F.B.); domenico.girelli@univr.it (D.G.); giulia.sartori.verona@gmail.com (G.S.); 3Department of Medicine, Allergy and Clinical Immunology Section, University of Verona and Azienda Ospedaliera Universitaria Integrata of Verona, 37134 Verona, Italy; marco.caminati@univr.it (M.C.); gianenrico.senna@univr.it (G.S.); 4INSERM, UMRS1158 Neurophysiologie Respiratoire Expérimentale et Clinique, AP-HP, Sorbonne Université, Groupe Hospitalier Pitié-Salpêtrière Charles Foix, Service des Explorations Fonctionnelles de la Respiration, de l’Exercice et de la Dyspnée du Département Médico-Universitaire «APPROCHES», 75013 Paris, France; pierantonio.laveneziana@aphp.fr; 5Respiratory Medicine Unit, University of Verona and Azienda Ospedaliera Universitaria Integrata of Verona, 37134 Verona, Italy

**Keywords:** cardiopulmonary exercise test, COVID-19, exercise ventilatory inefficiency, heart rate recovery, cardiovascular alterations

## Abstract

The cardiopulmonary exercise test (CPET) provides an objective assessment of ventilatory limitation, related to the exercise minute ventilation (V_E_) coupled to carbon dioxide output (V_CO2_) (V_E_/V_CO2_); high values of V_E_/V_CO2 slope_ define an exercise ventilatory inefficiency (EV*in*). In subjects recovered from hospitalised COVID-19, we explored the methodology of CPET in order to evaluate the presence of cardiopulmonary alterations. Our prospective study (RESPICOVID) has been proposed to evaluate pulmonary damage’s clinical impact in post-COVID subjects. In a subgroup of subjects (RESPICOVID2) without baseline confounders, we performed the CPET. According to the V_E_/V_CO2 slope_, subjects were divided into having EV*in* and exercise ventilatory efficiency (EV*ef*). Data concerning general variables, hospitalisation, lung function, and gas-analysis were also collected. The RESPICOVID2 enrolled 28 subjects, of whom 8 (29%) had EV*in*. As compared to subjects with EV*ef*, subjects with EV*in* showed a reduction in heart rate (HR) recovery. V_E_/V_CO2 slope_ was inversely correlated with HR recovery; this correlation was confirmed in a subgroup of older, non-smoking male subjects, regardless of the presence of arterial hypertension. More than one-fourth of subjects recovered from hospitalised COVID-19 have EV*in*. The relationship between EV*in* and HR recovery may represent a novel hallmark of post-COVID cardiopulmonary alterations.

## 1. Introduction

Shortly after discharge, survivors of COVID-19 present lung function alterations with reduction in diffusion capacity for carbon monoxide (DL_CO_) [1] and severe impairments in physical function during activities of daily living [2]. Few data are available about a comprehensive evaluation of COVID-19 clinical alterations during a more extended period.

The cardiopulmonary exercise test (CPET) provides an objective assessment of exercise capacity, adding physiological aspects that limit the individual’s performance [3]. In particular, the exercise minute ventilation (V_E_) relative to carbon dioxide output (V_CO2_) (V_E_/V_CO2_) shows complementary information about ventilatory limitation and ventilatory control [4,5]. During incremental exercise, the relationship between V_E_ and V_CO2_ may be plotted on a *y*-axis (V_E_) and *x*-axis (V_CO2_); the slope of this regression line (V_E_/V_CO2 slope_) may be considered an indicator of ventilatory efficiency [4,5]. Lower and upper limits of normal range of V_E_/V_CO2 slope_ are reported from approximately 21 to 31 [4,6,7]. High values of V_E_/V_CO2 slope_ define an exercise ventilatory inefficiency (EV*in*) [4,5]; this pathophysiological feature may explain the out-of-proportion breathlessness of patients with chronic obstructive pulmonary disease (COPD) [5]. Smokers with normal spirometry but with low values of DL_CO_ may have EV*in* [8].

In our pilot study, we explored the methodology of CPET to post-COVID subjects in order to evaluate the presence of cardiopulmonary alterations.

## 2. Materials and Methods

A dedicated outpatient clinic has been organised at our tertiary hospital enrolling all adult subjects previously hospitalised for interstitial pneumonia due to COVID-19, with or without respiratory failure. The prospective RESPICOVID study has been designed to evaluate the prevalence and the clinical impact of pulmonary damage in subjects recovered from COVID-19. In a subgroup of subjects (RESPICOVID2) a CPET has been performed. The study protocol was approved by the local Ethics Committee (no. 2785CESC), according to the Good Clinical Practice recommendations and the requirements of the Declaration of Helsinki. Written informed consent was obtained from all participants.

### 2.1. Inclusion Criteria

All consecutive patients discharged were considered.

### 2.2. Exclusion Criteria

The study has not considered subjects with the following criteria: (a) age > 65 years; (b) all concomitant previous respiratory or non-respiratory diseases; (c) chronic respiratory failure or need for oxygen-therapy under exertion; (d) moderate obesity defined by a body mass index (BMI) ≥ 35 kg/m^2^; (e) inability to perform functional tests; (f) inability to perform a CPET with a peak respiratory exchange ratio (RER) < 1.05 (to exclude poor motivation). Among chronic diseases, only stable arterial hypertension was accepted.

### 2.3. Measurements

All measures were collected prospectively beginning on 17 July 2020, after more than five months from subjects’ discharge (mean time 169 days, standard deviation (SD) 28 days). We recorded demographic and anthropometric variables, data concerning the hospitalisation, clinical symptoms, and gas-analysis.

Lung function and CPET procedures were performed according to international recommendations [3,9]. A flow-sensing spirometer connected to a computer for data analysis (Jaeger MasterScreen PFT System) was used to measure lung function. Forced vital capacity (FVC), forced expiratory volume in the first second (FEV_1_), total lung capacity (TLC), and inspiratory capacity (IC) were recorded. FEV_1_/FVC ratio and IC/TLC ratio were taken as the index of airflow obstruction and resting hyperinflation, respectively. Diffusion capacity for carbon monoxide (DL_CO_) was measured by the single breath method. FEV_1_, TLC, and DL_CO_ were expressed as percentages of the predicted values [10,11]. For the CPET, according to the ATS/ACCP Statement [3], we used a cycle ergometer (Cosmed, Milan, Italy) with a ramp protocol of 10 to 25 watts increment every minute and based on the predicted peak power output, in order to achieve an exercise time between 8 and 12 min. Subjects were asked to avoid caffeine, alcohol, cigarettes, and strenuous exercise 24 h before the day of testing; to eat a light breakfast; and to avoid eating for the 2 h before the test. Subjects suspended β-blockers before testing, but they could take their current antihypertensive therapies. During the test, subjects were asked to maintain a pedal frequency of 65 per minute and were continuously monitored [3]. Patients were continuously monitored with a 12-lead electrocardiogram (ECG) and a pulse oximeter; blood pressure was measured every two minutes. Stopping criteria consisted of symptoms, such as unsustainable dyspnoea, leg fatigue or chest pain, a significant ST-segment depression at ECG, or a drop in systolic blood pressure or oxygen saturation ≤84% [3]. Oxygen uptake (V_O2_) at the peak was expressed in mL/kg/min. The ventilatory response during exercise was expressed as a linear regression function by plotting minute ventilation (V_E_) against carbon dioxide production (V_CO2_) obtained every 10 s, excluding data above the ventilatory compensation point, and the slope (V_E_/V_CO2 slope_) and Y-intercept (V_E_/V_CO2 intercept_) values were obtained from the regression line. We used the regression equation of V_E_/V_CO2 slope_ for healthy subjects, according to Sun et al. [6], considering three standard deviations as the upper limit. Then, we considered subjects having a normal range of V_E_/V_CO2 slope_ (exercise ventilatory efficiency-EV*ef*) and subjects with over the upper limit of V_E_/V_CO2 slope_ (EV*in*). The cardiovascular response to exercise was expressed by the oxygen pulse (O_2_ pulse), the double product (DP) reserve, and the heart rate (HR) recovery, considering the value of heart rate measured after 1 min of exercise stops. At the end of the exercise, dyspnoea and leg fatigue were measured by a Borg 6–20 perceived exertion rate (RPE) scale [12]. Reasons for considering a maximal test were (a) a plateau of the V_O2_ more than 20 s; (b) a RER >1.15; (c) a rate of perceived exertion >18 on the Borg RPE scale [3].

As a measure of physical tolerance, walking capacity was assessed by the 6 min walking distance (6MWD) and performed according to the recommended guidelines [13]; the better of two consecutive tests was considered for the analysis. The reference equation for healthy adults was also used [14].

The Italian version of the International Physical Activity Questionnaire (IPAQ) was administered to measure the physical activity of the subjects in the last seven days, deriving three levels of metabolic equivalent of task (METs): inactive, minimally active, and health-enhancing physical activity (HEPA) active [15].

A preliminary Shapiro–Wilk test was performed. Data are reported as percentages for categorical variables, as mean (SD) or median [first quartile; third quartile] for continuous variables with a normal or non-normal distribution. Categorical variables were compared by the χ2 test or the Fisher exact test, while continuous variables were assessed by the independent t-test or the non-parametric Mann–Whitney test. Pearson (r) and Spearman (ρ) correlations have been carried out between parametric variables. The area under a receiver operating characteristic curve (AUC) measured the diagnostic discrimination property of significant predicting ventilatory inefficiency. All analyses were performed using IBM SPSS, version 17.0 (IBM Corp., Armonk, NY, USA), with *p*-values of <0.05 considered statistically significant.

## 3. Results

The RESPICOVID study enrolled 130 subjects, but according to the selective criteria for the RESPICOVID2, defined to avoid baseline bias influencing the ventilatory response to exercise, we performed the CPET in 28 subjects. All subjects performed a maximal exercise test, and 8 out of 28 (29%) had EV*in*. As compared to subjects with EV*ef*, subjects with EV*in* showed a reduction in HR recovery and V_E_/V_CO2 intercept_, with an increase by definition of the V_E_/V_CO2 slope_ and vigorous METs (Table 1). V_E_/V_CO2 slope_ was inversely correlated with HR recovery (r −0.537, *p* = 0.003) (Figure 1); this correlation was confirmed in a subgroup of older subjects (age > 55 y, N = 14, ρ −0.611, *p* = 0.020), males (N = 22, r −0.543, *p* = 0.009), non-smokers (N = 19, r −0.611, *p* = 0.005), regardless of the presence of arterial hypertension (yes, N = 9, r −0.669, *p* = 0.049; no, N = 19, r −0.487, *p* = 0.034).

The accuracy analysis of HR recovery showed a significant predictive discrimination (AUC, 0.767; standard error, 0.10; 95% confidence interval, 0.568 to 0.966; *p* = 0.028) with the best cutoff of 22 beats/minute (0.750 and 0.727 in the sensitivity and specificity evaluation) (Figure 2).

Subjects with arterial hypertension were treated with ACE inhibitors (N = 5, 18%), β-blockers (N = 4, 14%), and Ca^2+^ antagonist (N= 3, 11%) with no differences between subjects with EV*ef* and EV*in*.

## 4. Discussion

Our pilot study is the first evaluating, in survivors of COVID-19 pneumonia, the role of CPET variables during an extended follow-up after hospital discharge. Although our considered subjects had a normal lung function and a preserved maximal exercise capacity, surprisingly, more than one-fourth had an EV*in*, which is a determinant of HR recovery, especially older male non-smokers, regardless of the presence of arterial hypertension.

In smokers with normal spirometry but low values of DL_CO_, EV*in* may be present [8] as well as [8] an impaired peripheral endothelial function [16]. In the context of alveolar-capillary membrane damage, decrements in DL_CO_ may be more likely related to pulmonary microvascular abnormalities than impaired gas distribution [8]. We may hypothesise a similar mechanism in our post-COVID subjects, in which we observe five months from discharge a selective lung function impairment in DL_CO_ reduction.

Ventilatory inefficiency in the healthy population is not a common occurrence. The normal upper limit of V_E_/V_CO2 slope_ is 31 [6,7]. Variables related to age and sex [6,17], such as chronic pulmonary and cardiovascular conditions, may influence the exercise ventilatory efficiency [3,4]; however, there is no concrete evidence that the fitness level has an impact on exercise ventilation [18,19]. In addition, regular endurance training may improve exercise ventilatory efficiency (potentially with a reduction in V_E_/V_CO2 slope_) by reducing peripheral chemoreceptor sensitivity [18]. In our sample, subjects having EV*in* had a coexisting presence of higher values of 6MWD, workload, and V_O2_ at peak, signs of a higher aerobic capacity. The levels of vigorous weekly METs were higher compared to subjects with EV*ef*; however, this was not correlated to V_E_/V_CO2 slope_ or HR recovery (data not shown). Interestingly, ventilatory efficiency is not related to residual lung function limitations and specific treatment during the COVID-19 hospitalisation. Although speculative, our findings of exercise ventilatory alterations in post-hospitalised subjects, evaluated without baseline bias, seems to be specific and COVID-related.

HR recovery represents a marker of cardiac autonomic dysfunction and a predictor of mortality in adults without heart disease history [20]. In COPD patients, HR recovery is associated with endothelial dysfunction, representing peripheral impairment [21]. Moreover, EV*in* in COPD is a predictor of the delay of HR recovery [22]. Although our post-COVID survivors do not have an airways obstruction as in COPD, recent reports highlighted frequent extrapulmonary manifestations, especially involving the cardiovascular system (myocardial dysfunction, arrhythmia, and acute coronary syndromes), attributed to virus-mediated endothelial-cell damage [23]. Our findings on EV*in* and HR recovery could therefore represent a novel hallmark of post-COVID cardiopulmonary alterations. In these subjects, also with low cardiovascular risk (non-smokers without arterial hypertension), an in-depth assessment of exercise-induced ventilatory and cardiovascular parameters by CPET allows the identification (or monitoring) of early specific alterations. Further studies are needed to evaluate the progressive persistence and the prognostic role of these alterations.

As a limitation, we acknowledge the small number of subjects included, related to selective criteria considering younger subjects without previous diseases. Moreover, we lack data concerning the residual organic pulmonary damage (by lung ultrasound or thorax computed tomography scan) with indirect signs of pulmonary hypertension (by echocardiography).

## 5. Conclusions

More than one-fourth of post-COVID subjects present an exercise ventilatory inefficiency related to lower heart rate recovery; this aspect may be a sign of systemic alterations present in these subjects. Further studies in a very large cohort of subjects need to confirm our finding. In the future, it may be interesting to apply the methodology of CPET in elderly patients with or without coexisting diseases to evaluate the impact of COVID-19 on single chronic conditions.

We suggest CPET as a potentially useful tool for identifying ventilatory and cardiovascular alterations in subjects recovered from COVID-19. Moreover, CPET may be useful as a monitoring system for exercise capacity and cardio-ventilatory limitations in subjects admitted to a rehabilitation program.

## Figures and Tables

**Figure 1 diagnostics-11-00507-f001:**
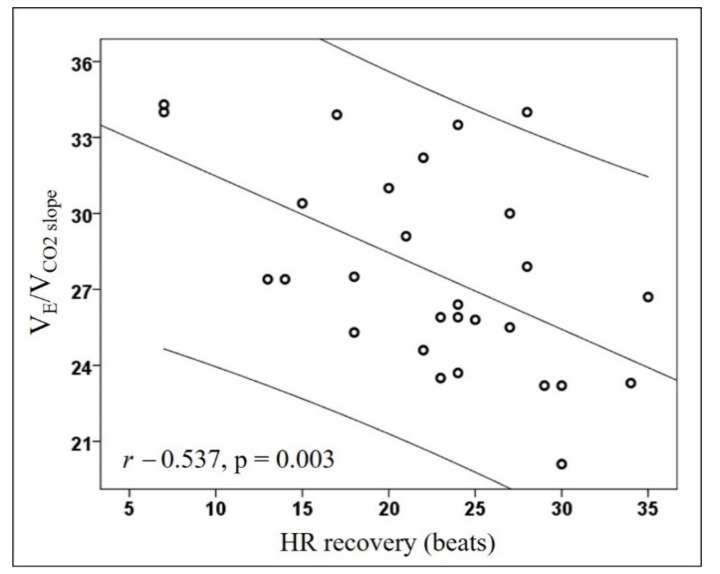
Scatterplot between V_E_/V_CO2_ slope and HR recovery. Lines represent the regression with the 95% confidence intervals. Abbreviations: V_E_/V_CO2 slope_ represents the slope of minute ventilation-V_E_ to carbon dioxide output-V_CO2_ ratio; HR, heart rate.

**Figure 2 diagnostics-11-00507-f002:**
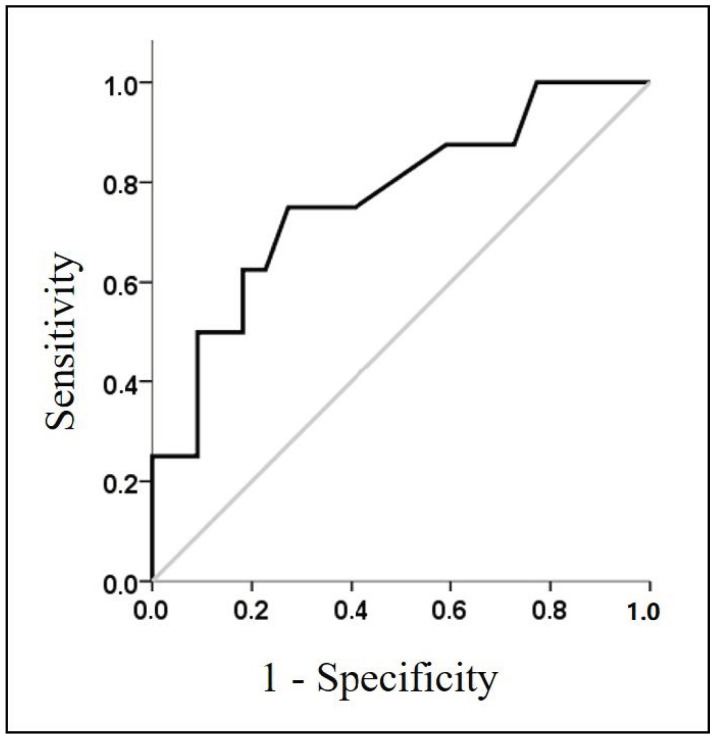
Receiver operating curve of HR recovery, performed on subjects with ventilatory inefficiency as test variable. Gray line represents a diagonal of reference.

**Table 1 diagnostics-11-00507-t001:** General and CPET-related variables.

Variables	All Subjects N = 28	Subjects with EV*ef* N = 20	Subjects with EV*in* N = 8	*p*-Value
Age, y	55.3 [52.3; 61.9]	55.1 [53.6; 59.2]	58.4 [48.7; 63.7]	0.576
Male, n (%)	22 (79)	15 (75)	7 (87)	0.640
BMI, kg/m^2^	25.9 ± 3.4	25.8 ± 3.2	26.2 ± 4.1	0.765
FFMI, kg/m^2^	19 ± 2.2	19 ± 2.4	18.9 ± 1.9	0.907
Smoking habit, no/current or former, n (%)	19 (68)/9 (32)	13 (65)/7 (35)	6 (75)/2 (25)	>0.999
Arterial hypertension, yes, n (%)	9 (32)	6 (30)	3 (37)	>0.999
FEV_1_, % pred.	118.1 ± 13.6	118.9 ± 14	116.1 ± 13.1	0.629
FEV_1_/FVC, %	101 ± 6.1	101.3 ± 6.5	100.2 ± 5.6	0.679
TLC, % predicted	104.2 ± 12	105.6 ± 13	100.6 ± 8.8	0.333
IC/TLC at rest	0.50 ± 0.08	0.49 ± 0.09	0.51 ± 0.06	0.649
DL_CO_, % predicted	89.9 ± 13.5	90.2 ± 13.9	89.4 ± 13.3	0.888
PaO_2_/FiO_2_	484.6 ± 37.6	477.7 ± 40.0	500 ± 27.7	0.169
PaCO_2_, mmHg	38.2 ± 3	38.4 ± 2.6	37.8 ± 3.9	0.699
6MWD, meters	604.5 ± 67.1	598.2 ± 56.1	620.4 ± 91.9	0.440
6MWD, % predicted	103 ± 15.2	101.8 ± 15.4	106.2 ± 15.1	0.502
IPAQ (inactive, minimally active, HEPA active), n (%)	4(14)/15(54)/9(32)	4(20)/12(60)/4(20)	0(0)/3(37)/5(63)	0.101
METs, vigorous	0 [0; 1320]	0 [0; 420]	1520 [120; 6120]	0.018
METs, total	1912.5 [1015.5; 3410.2]	1372 [838.5; 2497]	2805 [1698.7; 10,865.5]	0.053
Workload, watts	187.7 ± 64	181.7 ± 56	202.7 ± 83.4	0.444
RER	1.19 [1.11; 1.25]	1.20 [1.13; 1.27]	1.12 [1.10; 1.20]	0.062
V_O2_ at peak, mL/kg/min	29.2 ± 8.3	27.6 ± 5.2	32.9 ± 13.1	0.137
V_O2_ at AT, mL/kg/min	17.6 [15.9; 22.4]	17.6 [16.2; 20.4]	20 [13.5; 29.7]	0.684
O_2_ pulse at rest, mL/beat/min	7.3 [5.8; 7.8]	7.5 [6.9; 7.9]	6.2 [5.4; 7.5]	0.169
O_2_ pulse at peak, mL/beat/min	14.5 ± 3.9	13.8 ± 3.8	16.1 ± 4	0.168
PET_CO2_ change ^1^	3.1 ± 4.4	3.7 ± 4.7	1.5 ± 3.6	0.235
V_E_ at rest	16.9 ± 4.1	16.6 ± 4.4	17.8 ± 3.2	0.470
V_E_ at peak	95.2 ± 33.4	89.2 ± 27.3	110.4 ± 43.9	0.131
RR at rest, bpm	15.9 ± 3.5	15.7 ± 3.8	16.5 ± 2.8	0.637
RR at peak, bpm	36.4 ± 8.9	34.4 ± 7.4	41.4 ± 10.8	0.057
V_O2_/Watts, mL/min/watts	11.8 [11.5; 12.6]	11.8 [11.4; 12.3]	12.2 [11.6; 13.9]	0.263
V_E_/V_CO2 slope_	27.7 ± 3.9	25.6 ± 2.3	32.9 ± 1.5	<0.001
V_E_/V_CO2_ at AT	28.9 ± 2.9	28.2 ± 2.7	30.5 ± 3	0.066
V_E_/V_CO2 intercept_	2.35 [0.12; 5.37]	3.65 [1.75; 5.87]	−1.10 [−3.52; 0.57]	<0.001
HR/V_O2_ slope, L^−1^	44.5 [38.2; 70]	47.2 [39.8; 74.9]	37.2 [34.4; 59.1]	0.060
Brething reserve, %	36.5 ± 14.7	39.8 ± 13	28.1 ± 16.3	0.054
V_D_/V_T_	0.26 ± 0.02	0.26 ± 0.02	0.27 ± 0.02	0.151
SBP at rest, mmHg	120 [115; 125]	120 [116.2; 125]	120 [111.2; 125]	0.853
SBP at peak, mmHg	183.7 ± 18.4	185.7 ± 19.1	178.7 ± 16.4	0.373
DBP at rest, mmHg	80 [70; 80]	80 [70; 83.7]	80 [70; 80]	0.625
DBP at peak, mmHg	95.4 ± 10.3	94.5 ± 9.9	97.5 ± 11.3	0.495
HR at rest, beats/min	69.7 ± 8.9	70.1 ± 10.1	68.9 ± 5.2	0.749
HR at peak, beats/min	156.6 ± 18.7	158.4 ± 17.6	152.1 ± 21.7	0.429
HR recovery, beats/min	22.4 ± 7	24.4 ± 5.8	17.5 ± 7.6	0.015
DP reserve	21060 [16,515; 22,013]	21030 [17,647; 22,445]	21,060 [12,630; 21,952]	0.647
RPE_dyspnea_, score	16.2 ± 2.6	16 ± 2.4	16.8 ± 3	0.430
RPE_fatigue_, score	17.5 [16.2; 19]	17.5 [16.2; 19]	18 [15.5; 19.7]	0.796
Variables related to COVID-19 hospitalisation				
PaO_2_/FiO_2_ ^2^ ≤ 300, n (%)	13 (46)	9 (45)	4 (50)	>0.999
ICU/medical ward ^3^, n (%)	5 (18)/23 (82)	2 (10)/18 (90)	3 (37)/5 (63)	0.123
Length of stay, d	5.9 [4.2; 10.5]	5.9 [4.2; 9.7]	5.5 [3.6; 21.9]	0.779
Pulmonary embolism, n (%)	2 (7.1)	1 (5)	1 (12.5)	0.497
Oxygen-therapy, n (%)	16 (57)	10 (50)	6 (75)	0.401
Ventilatory support ^4^, n (%)	10 (36)	6 (30)	4 (50)	0.400
Lopinavir/ritonavir, n (%)	22 (79)	16 (80)	6 (75)	>0.999
Hydroxychloroquine, n (%)	26 (93)	18 (90)	8 (100)	>0.999
Antibiotics, n (%)	9 (32)	7 (35)	2 (25)	>0.999
Tocilizumab, n (%)	8 (29)	5 (25)	3 (37)	0.651
Steroids, n (%)	13 (46)	8 (40)	5 (62)	0.410
Prophylactic LMWH, n (%)	8 (29)	6 (30)	2 (25)	>0.999

Data are shown as the number of subjects (%), means ± SD or medians [first quartile; third quartile]. In bold, significant variables. Abbreviations: EV*ef* and EV*in* define exercise ventilatory efficiency and inefficiency, respectively; BMI, body mass index; FFMI, fat-free mass index, calculated as FFM/height squared; FEV_1_, forced expiratory volume at 1st second; FVC, forced vital capacity; TLC, total lung capacity; IC, inspiratory capacity; DL_CO_, diffusion capacity for carbon monoxide; PaO_2_, arterial partial oxygen pressure; FiO_2_, fraction of inspired oxygen; PaCO_2_, arterial partial carbon dioxide pressure; 6MWD, 6-min walked distance; IPAQ, international physical activity questionnaire; HEPA, health-enhancing physical activity; METs, metabolic equivalent of task; RER, respiratory exchange ratio; V_O2_, oxygen uptake; PET_CO2_, end-tidal pressure of CO_2_; V_E_, minute ventilation; RR, respiratory rate; V_E_/V_CO2 slope_, the slope of V_E_ to carbon dioxide output-V_CO2_ ratio; AT, anaerobic threshold; V_E_/V_CO2 intercept_, point of intercept of V_E_ to carbon dioxide output-V_CO2_ ratio; HR, heart rate; V_D_, dead space; V_T_, tidal volume; SBP and DBP, systolic and diastolic blood pressure, respectively; DP, double product; RPE, rate of perceived exertion; ICU, intensive care unit. ^1^ calculated as peak PET_CO2_ minus at rest PET_CO2_; ^2^ at hospital admission; ^3^ unit of admission; ^4^ include subjects treated with continuous positive airway pressure (CPAP) and pressure support ventilation (PSV).

## Data Availability

The data presented in this study are available on request from the corresponding author.

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
