# Peer review of "Importance of Cardiopulmonary Exercise Testing amongst Subjects Recovering from COVID-19"

_diagnostics, 2021, doi:10.3390/diagnostics11030507_

Round 1

Reviewer 1 Report

Very good and actual article.

Author Response

Q) Very good and actual article.

A) We thank the Reviewer for his/her positive comments.

Reviewer 2 Report

What may add the cardiopulmonary exercise test in the evaluation of subjects recovered from hospitalized COVID-19? A pilot study

May thanks for submitting an informative and topical study. Although it is important to understand the impact of COVID-19 on the cardiopulmonary system, there are some important considerations required for this manuscript. Please see below for more detailed information.

Perhaps a better title would be

Cardiopulmonary Exercise Testing – An objective assessment of ventilatory limitation amongst individuals recovering from COVID-19 – A pilot study.

Or

Importance of cardiopulmonary exercise testing amongst individuals recovering from COVID-19.

Abstract

The authors have a clear aim, to provide an objective assessment of ventilatory limitation in individuals recovered from hospitalized COVID-19. This is an important topic as currently it is suggested that all individuals recovering from COVID-19 should not participate in exercise. There are, a paucity of studies within this area and it is important to understand why and which individuals have problems and should be monitored carefully.

The study sample size is relatively small, but nevertheless the authors have shown that some subjects have EVin, compared to others.

Introduction

The authors not the paucity of information surrounding physical function and activities of daily living and the importance of CPET testing. The authors also discuss VE/VCO2 slope along with exercise ventilatory insufficiency in individuals with COPD and smokers. It would be advantageous for the reader, if the authors could discuss the numbers associated with this so that the reader has an understanding of the what the normal slope and exercise ventilatory sufficiency would be in healthy individuals.

Materials and Methods

I would ask the authors to remove the word we…. In the first paragraph please review and include an inclusion and exclusion criteria as the way the article is currently written I am unsure of whether this is the inclusion or exclusion criteria for the study.

For the CPET test, were subjects given the option of either 10 or 25W increments? How did you decide on the increments for each subject? Why did you not use a standard 15 or 25W per minute increment?

The authors state that they used the 6-20 scale for both dyspnea and also leg fatigue. It is my understanding that the Borg 6-20 scale is not validated for this and I would ask the authors to include references to back this up….

Results

The authors mention that individuals completed a maximal exercise test – how and what criteria was used to ensure that this was the case for all subjects. In the results, I would expect to see RPE, RER at peak exercise alongside peak BP, HR, VO2, Power output and VE.  It would be also of interest to the reader to understand if any of the subject had been on a. ventilator during the hospital stay.

It is of interest to note that only 28 subjects were tested via CPET. It would be good to understand why the other 102 were not tested and how this may impact future testing.

I wonder if there is an error in line 118 of the results and whether after N=14 should read r—0.611.

The results are interesting and it appears that all individuals reached 95% of age predicted HR max during their CPET test.

It would have been interesting to test FEV1, FEV1/FVC during the 20 minutes following CPET testing as this may have demonstrated alterations in breathing over this period. It is also likely that cardiac output may also be reduced in the subject with EVin.

Discussion

The discussion is very short and could be longer, discussing the merits of CPET testing in this group of subjects. The discussion needs to explain and critically evaluate the findings and that to other conditions and to control subjects. It would have been good if the authors had have included a control group who had not suffered from COVID-19 and were aged and gender matched controls. This would better aid the understanding of the impact of COVID-19 on ventiltory capacity/cardiovascular insufficiency.

The conclusion could also gain a little more depth, rather than just one sentence.  

Author Response

Q) Perhaps a better title would be

Cardiopulmonary Exercise Testing – An objective assessment of ventilatory limitation amongst individuals recovering from COVID-19 – A pilot study.

or

Importance of cardiopulmonary exercise testing amongst individuals recovering from COVID-19.

A) We thank the Reviewer for his/her positive suggestion. We have modified the title, accordingly.

Q) The authors not the paucity of information surrounding physical function and activities of daily living and the importance of CPET testing. The authors also discuss VE/VCO2 slope along with exercise ventilatory insufficiency in individuals with COPD and smokers. It would be advantageous for the reader, if the authors could discuss the numbers associated with this so that the reader has an understanding of the what the normal slope and exercise ventilatory sufficiency would be in healthy individuals.

A) We thank the Reviewer for his/her positive suggestions. We have added in the introduction some aspects related to the normal range of VE/VCO2 slope.

Q) I would ask the authors to remove the word we…. In the first paragraph please review and include an inclusion and exclusion criteria as the way the article is currently written I am unsure of whether this is the inclusion or exclusion criteria for the study.

A) We thank the Reviewer for his/her suggestions. We have modified the text of Material and Methods accordingly.

Q) For the CPET test, were subjects given the option of either 10 or 25W increments? How did you decide on the increments for each subject? Why did you not use a standard 15 or 25W per minute increment?

A) We thank the Reviewer for his/her suggestions. We used the predicted peak power output, such as described by the ATS/ACCP Statement for CPET; the choice of 10 to 25 watts every minute is defined to obtain an incremental phase from 8 to 12 minutes.

Q) The authors state that they used the 6-20 scale for both dyspnea and also leg fatigue. It is my understanding that the Borg 6-20 scale is not validated for this and I would ask the authors to include references to back this up….

A) We thank the Reviewer for his/her suggestion. We have added in the modified version of the manuscript the reference for the Borg 6-20 perceived exertion rate (RPE) scale. This scale is commonly used for the exercise evaluation and prescription (Noble BJ. Med Sci Sports Exerc. 1982), similarly to Borg 0-10 scale (Arney BE et al. Int J Sports Physiol Perform. 2019).

Q) The authors mention that individuals completed a maximal exercise test – how and what criteria was used to ensure that this was the case for all subjects.

A) We thank the Reviewer for his/her positive suggestions. In the revised version of the manuscript, we have added a sentence about reasons to consider a maximal test.

Q) In the results, I would expect to see RPE, RER at peak exercise alongside peak BP, HR, VO2, Power output and VE.

A) We thank the Reviewer for his/her suggestions. We have reported in the first version of the manuscript data concerning the RPE, HR and VO2 at peak. Now we have added data concerning RER at peak and VE.

Q) It would be also of interest to the reader to understand if any of the subject had been on a. ventilator during the hospital stay.

A) We thank the Reviewer for his/her suggestion. We have added in the manuscript data about treatments concerning also oxygen therapy and ventilatory support.

Q) It is of interest to note that only 28 subjects were tested via CPET. It would be good to understand why the other 102 were not tested and how this may impact future testing.

A) We thank the Reviewer for his/her suggestions. We have added a sentence about reasons for exclusion criteria; moreover, we have added a future possibility for patients with chronic disease.

Q) I wonder if there is an error in line 118 of the results and whether after N=14 should read r—0.611.

A) We thank the Reviewer for his/her comment. The phrase is correct; we used the Spearman rho (ρ) due to the age's non-normal distribution.

Q) It would have been interesting to test FEV1, FEV1/FVC during the 20 minutes following CPET testing as this may have demonstrated alterations in breathing over this period. It is also likely that cardiac output may also be reduced in the subject with EVin.

A) We thank the Reviewer for his/her comment. We agree with this possibility suggested by the Reviewer, but we are unable to have this variables.

Q) The discussion is very short and could be longer, discussing the merits of CPET testing in this group of subjects. The discussion needs to explain and critically evaluate the findings and that to other conditions and to control subjects.

A) We thank the Reviewer for his/her positive suggestions. We have added some aspects related to normal subjects in the discussion, compared this with our study sample. We hope it will be appreciated by the Reviewer.

Q) It would have been good if the authors had have included a control group who had not suffered from COVID-19 and were aged and gender matched controls. This would better aid the understanding of the impact of COVID-19 on ventiltory capacity/cardiovascular insufficiency.

A) We thank the Reviewer and we agree for his/her suggestion. Actually, in the future, we will organize, such as proposed from the Reviewer, a control group, healthy not COVID-19, age and gender-matched, enrolled from workers in our hospital.

Q) The conclusion could also gain a little more depth, rather than just one sentence.  

A) We thank the Reviewer for his/her suggestions. We have added a sentence in conclusion concerning the findings.

Reviewer 3 Report

In this paper Dorelli and colleagues present an interesting study involving 28 otherwise healthy patients who were previously hospitalized for symptomatic COVID-19 pneumonia. Patients underwent a follow-up cardiopulmonary exercise test (CPET). 8 out of 28 patients (29%) showed exercise ventilatory inefficiency (EVin). Such patients showed also a lower heart rate recovery (HRR) as compared to patients with normal exercise ventilatory efficiency (EVef). I would like to congratulate the authors for organizing such a dedicated outpatient clinic. The topic is relevant and may be of strong interest for the readers. Originality is high. However some issue should be addressed. First of all the manuscript require extensive editing of English language (a native english speaker would be welcomed) and style. Results and conclusions are too gaunt. Some other important issues need to be addressed.

A major limit is the lack of any correlation between Evef and radiological and cardiological (mainly by echocardiography) informations. Chest CTA would be usefull to better characterize patients with Evef, as well as echocardiography could evaluate indirect signs of pulmonary hypertension. Also baseline characteristics are scarce. Patients who needed oxygen, C-PAP, pharmacological therapy adopted (including also heparin) etc during hospitalization should be provided. It is not clear why results and discussion are focused on HRR. This can also be calculated by a simple exercise ECG test, so the advocated CPET utility can be questioned. However a clear definition of HRR is lacking. VE/VCO2 cut-off adopted should be better explained. Some important CPET parameters have not been reported. These include: HR/VO2 (chronotropic response to exercise), VO2/Work for cardiocirculatory efficiency, breathing reserve, death space (VD/VT) and anaerobic threshold. 

Furthermore it is surprising that patients with Evin have such high values of workload (mean 202 watts) and VO2 at peak (mean 32.9 ml/Kg/min). 

In the methods it should be reported if chronotropic active drugs were suspended before CPET.

The scatterplot between VE/CO2 and HRR formally shows statistical significancy but with wide dispersion. Maybe a ROC analysis with VE/VCO2 slope as categorical variable and HRR as continuos variable may reinforce this finding. Other covariates should be explored in the same way. 

A potential role of CPET in this setting is also to monitoring exercise capacity (even after rehabilitation program) and to identify the source of functional impairment. This should be discussed.

Author Response

In this paper Dorelli and colleagues present an interesting study involving 28 otherwise healthy patients who were previously hospitalized for symptomatic COVID-19 pneumonia. Patients underwent a follow-up cardiopulmonary exercise test (CPET). 8 out of 28 patients (29%) showed exercise ventilatory inefficiency (EVin). Such patients showed also a lower heart rate recovery (HRR) as compared to patients with normal exercise ventilatory efficiency (EVef). I would like to congratulate the authors for organizing such a dedicated outpatient clinic. The topic is relevant and may be of strong interest for the readers. Originality is high.

A) We thank the Reviewer for his/her positive comments.

Q) First of all the manuscript require extensive editing of English language (a native english speaker would be welcomed) and style. Results and conclusions are too gaunt. Some other important issues need to be addressed.

A) We thank the Reviewer for his/her suggestion. We have performed an English revision by a native speaker.

Q) A major limit is the lack of any correlation between Evef and radiological and cardiological (mainly by echocardiography) informations. Chest CTA would be usefull to better characterize patients with Evef, as well as echocardiography could evaluate indirect signs of pulmonary hypertension. Also baseline characteristics are scarce.

A) We thank the Reviewer for his/her positive suggestion. However, in our organization of COVID-19 outpatients, we cannot perform a chest CT scan and echocardiography easily. For this aspect we have not added these variables in the protocol. In the revised version of the manuscript, we have added it as a limitation.

Q) Patients who needed oxygen, C-PAP, pharmacological therapy adopted (including also heparin) etc during hospitalization should be provided.

A) We thank the Reviewer for his/her positive suggestions. In the revised version, we have added more variables concerning treatment during hospitalization.

Q) It is not clear why results and discussion are focused on HRR. This can also be calculated by a simple exercise ECG test, so the advocated CPET utility can be questioned. However, a clear definition of HRR is lacking.

A) We thank the Reviewer for his/her suggestion. We have completed the sentence concerning the measure of HRR. In the discussion, we have added a paragraph concerning the ventilatory efficiency.

Q) VE/VCO2 cut-off adopted should be better explained.

A) We thank the Reviewer for his/her suggestions. We have added a sentence about upper limit of regression equation of Sun et al. (considering three standard deviations as upper limit) in the revised version of the manuscript.

Q) Some important CPET parameters have not been reported. These include: HR/VO2 (chronotropic response to exercise), VO2/Work for cardiocirculatory efficiency, breathing reserve, death space (VD/VT) and anaerobic threshold.

A) We thank the Reviewer for his/her suggestion. We have added variables describing HR/VO2 slope, breathing reserve, VD/VT and anaerobic threshold. Data about VO2/work was reported in the first version of the manuscript.

Q) Furthermore it is surprising that patients with Evin have such high values of workload (mean 202 watts) and VO2 at peak (mean 32.9 ml/Kg/min).

A) We thank the Reviewer for his/her suggestion. In the revised version of the manuscript, we have added a paragraph in discussion concerning the probable reason because subjects with ventilatory inefficiency may increase VO2 at peak and workload, similarly to data about 6MWD (now added).

Q) In the methods it should be reported if chronotropic active drugs were suspended before CPET.

A) We thank the Reviewer for his/her suggestion. We have added a sentence in the method concerning the withdraw of β-blockers.

Q) The scatterplot between VE/CO2 and HRR formally shows statistical significancy but with wide dispersion. Maybe a ROC analysis with VE/VCO2 slope as categorical variable and HRR as continuos variable may reinforce this finding. Other covariates should be explored in the same way.

A) We thank the Reviewer for his/her grateful suggestion. We have added in the results data concerning a ROC curve of HR recovery tested with ventilatory inefficiency. Moreover, a figure has been added (Figure 2).

Q) A potential role of CPET in this setting is also to monitoring exercise capacity (even after rehabilitation program) and to identify the source of functional impairment. This should be discussed.

A) We thank the Reviewer for his/her positive suggestions. We have added a sentence in conclusion concerning the potential role of CPET in monitoring COVID subjects.

Reviewer 4 Report

This study evaluates the useful of CPX test in the characterization of patients post Covid.

This pilot study shows that subjects with ventilatory inefficiency have a reduction in the heart rate recovery. The Authors conclude that this relation may represent a characteristic of the post Covid patients.

Main criticisms

a) the number of subjects with Covid is limited and should be certainly increased

b) the study lacks of a 6-min walking test that is usually used in patients with Covid in acute units. This test should be used acutely and in chronic phase and results should be included in the paper.

c) more important the study lacks of a good evaluation and definition of pulmonary damage via eco-fast and CT scan of thorax 

d) the use of beta blocking agents certainly has negative influence on heart rate thus I suggest to avoid this class of drugs before CPX tests

Author Response

Q) a) the number of subjects with Covid is limited and should be certainly increased

A) We thank and we agree with the Reviewer for his/her suggestion. However, we have explored the methodology of CPET to study healthy younger subjects without previous comorbidities. The exclusion of older patients with any comorbidities has reduced the study sample. We believe the message of the clinical utility of CPET in post-COVID patients needs to be suggested.

Q) b) the study lacks of a 6-min walking test that is usually used in patients with Covid in acute units. This test should be used acutely and in chronic phase and results should be included in the paper.

A) We thank the Reviewer for his/her suggestions. In the revised version, we have added values concerning 6MWD.

Q) c) more important the study lacks of a good evaluation and definition of pulmonary damage via eco-fast and CT scan of thorax

A) We thank and we agree with the Reviewer for his/her suggestions. However, in our organization of COVID-19 outpatients, we cannot perform a lung echography and a chest CT scan easily. We have added this aspect in the manuscript as a limitation.

Q) d) the use of beta blocking agents certainly has negative influence on heart rate thus I suggest to avoid this class of drugs before CPX tests

A) We thank the Reviewer for his/her positive suggestions. We have added a sentence in the method section concerning the CPET.

Round 2

Reviewer 3 Report

Authors have addressed at their best all the reviewer's comments. 

The quality of the paper seems now consistently increased. 

Reviewer 4 Report

In this form the paper appears partially improved. Limitations refer to the low number of Covid patients (only young) and lack of CT scan or chest eco to characterize the pulmonary alterations.